# Roles of Histone Acetylation Modifiers and Other Epigenetic Regulators in Vascular Calcification

**DOI:** 10.3390/ijms21093246

**Published:** 2020-05-04

**Authors:** Duk-Hwa Kwon, Juhee Ryu, Young-Kook Kim, Hyun Kook

**Affiliations:** 1Department of Pharmacology, Chonnam National University Medical School, Hwasun 58128, Korea; elio9359@hanmail.net (D.-H.K.); juheer12@gmail.com (J.R.); 2Department of Biochemistry, Chonnam National University Medical School, Hwasun 58128, Korea

**Keywords:** histone deacetylase, histone modifiers, epigenetic regulator, vascular calcification, vascular smooth muscle cells, post-translational modification

## Abstract

Vascular calcification (VC) is characterized by calcium deposition inside arteries and is closely associated with the morbidity and mortality of atherosclerosis, chronic kidney disease, diabetes, and other cardiovascular diseases (CVDs). VC is now widely known to be an active process occurring in vascular smooth muscle cells (VSMCs) involving multiple mechanisms and factors. These mechanisms share features with the process of bone formation, since the phenotype switching from the contractile to the osteochondrogenic phenotype also occurs in VSMCs during VC. In addition, VC can be regulated by epigenetic factors, including DNA methylation, histone modification, and noncoding RNAs. Although VC is commonly observed in patients with chronic kidney disease and CVD, specific drugs for VC have not been developed. Thus, discovering novel therapeutic targets may be necessary. In this review, we summarize the current experimental evidence regarding the role of epigenetic regulators including histone deacetylases and propose the therapeutic implication of these regulators in the treatment of VC.

## 1. Introduction

Vascular calcification (VC) is common in patients with chronic kidney disease, atherosclerosis, and diabetes. It is characterized by the accumulation of calcium phosphate products inside the vascular walls [1]. VC is an independent risk factor for cardiovascular diseases (CVDs) [2] and is highly associated with CVD mortality. VC can be categorized into two types: intimal calcification and medial calcification. Intimal calcification develops in the intimal layer of the vascular walls in patients with atherosclerosis, whereas medial calcification occurs in the medial layer of the vascular walls in patients with chronic kidney disease, diabetes, hypertension, and osteoporosis [3]. 

Previously, VC, an elevation of calcium phosphate crystals in the vascular walls, was considered to be the passive result of VSMC death. However, it is now known to be an active process involving various regulators of bone formation [4]. During the progression of VC, the expression of contractile markers such as smooth muscle 22 alpha (SM22α) and alpha smooth muscle actin (α-actin) are downregulated, whereas the expression of bone-related factors such as runt-related transcription factor 2 (RUNX2), msh homeobox (MSX), bone morphogenetic proteins (BMPs), and osteocalcin are upregulated [1]. Thus, smooth muscle cells (SMCs) undergo phenotype switching from the contractile phenotype to the osteochodrogenic phenotype [5]. Although VC could be treated with conventional therapeutics used for chronic kidney disease, CVD, and osteoporosis [3], it is still common in patients with chronic kidney disease and atherosclerosis. Thus, it is critical to find novel key regulators in the pathogenesis of VC and new therapeutic targets. 

Previously, epigenetic regulators were reported to be involved in the progression of diverse diseases such as cancer, autoimmune diseases, and neurological and psychological diseases [6] and showed potential as therapeutic targets [7]. *Epigenetic* refers to a heritable phenotype resulting from an alteration in gene expression without a change in DNA sequence [8]. Epigenetic regulations such as DNA methylation, histone modification, and noncoding RNAs (ncRNAs) [9] can be involved in the development of VC. In this review, we will focus on these three regulators with an emphasis on the role of histone modification.

## 2. Mechanisms of Vascular Calcification

VC is a complex and interactive process involving various calcification-related factors, apoptosis, mitochondrial dysfunction, and senescence [4,5]. Procalcifying factors such as BMP2 and RUNX2 can promote VC, whereas anticalcifying factors such as OPG and OPN may inhibit VC. Moreover, various signaling pathways such as BMP signaling and the Wnt/β-catenin pathway are involved in the development of VC [10]. Additionally, age-related factors including cell death and mitochondrial metabolism may affect VC.

### 2.1. Procalcifying Factors

The BMPs are members of the transforming growth factor-β (TGF-β) family that are reported to be involved in embryogenesis, organogenesis, and osteoblast differentiation [11]. Although there are more than 30 different types of BMPs [12], we will focus on BMP-2, which is well-known for its procalcifying properties. BMP-2 may promote VC by activating muscle segment homeobox2 (MSX2) and inhibiting matrix Gla protein (MGP). It may also promote apoptosis of vascular smooth muscle cells (VSMCs) [13]. Derwall et al. found that suppressing BMP-2 inhibited the formation of atheromas and VC in low-density lipoprotein receptor-deficient (LDLR^−/−^) mice [14]. On the other hand, VC was promoted in BMP-2 transgenic mice [15]. BMP signaling is activated when BMP-2 binds to type I and II BMP-2 receptor and phosphorylates SMAD (small mothers against decapentaplegic) 1/5/8. Phosphorylated SMAD 1/5/8 can enter the nucleus and further activate downstream calcification genes, such as RUNX2 and MSX2 [12,13]. Additionally, it was revealed that BMP-2 and MSX2 can activate the Wnt/β-catenin pathway and induce VSMC calcification [16,17]. The Wnt/β-catenin pathway is one of the major osteoinductive signaling pathways in VC [10]. WNT, a ligand protein, binds to the cell membrane receptors of the lipoprotein receptor-related protein 5/6 and Frizzled family and activates β-catenin. β-catenin translocates to the nucleus and activates downstream target genes, including calcification genes [18]. 

RUNX2 is a transcription factor involved in osteoblast differentiation and bone formation [19]. Although the expression of RUNX2 is low in normal vessels, it is highly expressed in calcified vessels, indicating that RUNX2 plays an important role in VC [20]. RUNX2 has been shown to induce calcification in VSMCs in vitro and was found to be critical in VSMC calcification induced by oxidative stress [21]. 

MSX2 is an essential transcription factor for bone formation and organogenesis [22,23]. MSX2 is upregulated in calcified arteries of diabetic mice, patients with diabetes, dyslipidemia, and vascular disease. On the other hand, downregulating MSX2 and MSX1 inhibits VC in diabetic LDLR mice [24]. 

Alkaline phosphatase (ALP) is a metalloenzyme and another key player of osteogenesis. It is widely expressed in various tissue, but is highly expressed in liver, kidney, and bone. Tissue-nonspecific ALP is activated by BMP2 and vitamin D agents, and this activation of ALP results in osteogenic transdifferentiation of VSMCs. ALP can act as a pyrophosphatase or can catalyze the hydrolysis of phosphomonoesters releasing inorganic phosphate (Pi). Elevated ALP levels alter osteoblasts and cause bone disease [25]. 

### 2.2. Anticalcifying Factors

Osteoprotegerin (OPG), which is a receptor for tumor necrosis factor, is reported to be involved in bone resorption [26]. OPG is expressed in diverse tissues such as heart, lung, and kidney and is involved in regulating the immune system [27]. OPG knockout (KO) mice exhibit severe osteoporosis and medial calcification [28], whereas transgenic mice overexpressing OPG show reduced osteoclast differentiation and enhanced bone mass [26]. The serum OPG level positively correlates with the severity of VC [29]. OPG acts as a decoy and interrupts the binding of receptor activator of NF-κB ligand (RANKL) to RANK, which ultimately inhibits osteoclast differentiation [27].

Osteopontin (OPN) is a phosphoprotein that is generally found in bone and teeth [30,31]. Although OPN is not commonly present in normal blood vessels, it is abundant in calcified atherosclerotic plaques and aortic valves [32,33,34]. Speer et al. revealed that MGP and OPN KO mice (MGP^−/−^, OPN^−/−^) develop more severe VC than MGP KO mice (MGP^−/−^, OPN^+/+^), which suggests that OPN has an inhibitory role in VC [35]. 

MGP, a vitamin K-dependent protein, is expressed in endothelial cells and VSMCs of normal blood vessels. Expression of MGP was reported to be downregulated in calcified vessels [36]. The severity of coronary artery calcification is negatively correlated with MGP levels in patients with suspected coronary artery diseases [37]. In addition, MGP levels are reduced in an in vitro VSMC calcification model [38], and MGP KO mice exhibit severe calcification [39]. These findings suggest that MGP acts as an inhibitor of VC. Moreover, MGP can interrupt BMP2 signaling by binding directly to BMP2, which can prevent BMP2 from binding to BMP receptors [39].

Fetuin-A, a glycoprotein expressed in adipose tissue and liver, is another key regulator in bone formation and resorption [40]. It inhibits VC by binding to calcium phosphate in the blood [41]. Fetuin-A KO mice demonstrate severe calcification in heart, kidney, skin, and tongue [42]. Coronary artery calcification scores are directly associated with fetuin-A levels [43,44]. In addition, administration of fetuin-A reduces calcification in vitro [45].

### 2.3. Other Factors Affecting VC

Apoptosis is another factor that directly influences VC. After cell death, the cell releases cellular DNA that precipitates calcium phosphate products and results in VC. Apoptosis occurs early in VSMC before VSMC calcification. Apoptotic bodies produced after cell death are high in calcium and are ultimately included in the formation of calcium phosphate crystals [46]. Suppression of apoptosis through caspase inhibitors reduces calcification and vesicles including calcium [47]. 

The phenotype of VSMCs can be influenced by changes in mitochondrial metabolism such as increased proliferation in pulmonary artery SMCs [48]. Mitochondrial dysfunction can contribute to the development of atherosclerosis [49]. Mitochondrial damage results in reduced adenosine triphosphate production and impaired respiratory chain function, which ultimately causes cellular malfunction and initiate atherosclerosis, a progressive disease that may lead to VC. Mitochondrial dysfunction also induces VC through the apoptotic pathway by promoting casapase-9, which releases cytochrome c [50]. 

VC was reported to be associated with aging and VSMC senescence [51]. Senescent cells release cytokines, proteases, and growth factors and stop mitosis. Osteogenic markers such as RUNX2, collagen 1, and ALP are increased in senescent VSMC cells in vitro, and medial calcification and upregulated RUNX2 levels are observed in the in vivo aging mouse model [52]. It is well known that senescence-related VC also increases microvesicles, a subset of extracellular vesicles. Microvesicles carry calcium, calcification nucleation-related proteins, and calcium-binding annexins and promote osteoblast transformation of VSMCs to induce VC [53] 

Endothelial cell can transform into mesenchymal stem cells, multipotent cells, by undergoing endothelial-to-mesenchymal transition (EndoMT). When mesenchymal stem cells are exposed to stimuli, they can differentiate into fibroblast, adipocyte, osteoblast, and chondrocyte [54]. Recent studies reported that TGF-β or BMP2-stimulated valvular endothelial cells undergo EndoMT process. These TGF-β or BMP2-stimulated valvular endothelial cells transform into osteoblast-like cells by increasing ALP expression during EndoMT, which eventually result in VC [55]. In contrast, suppression of EndoMT in valvular endothelial cells can inhibit VC [56].

## 3. Epigenetic Regulation in Vascular Smooth Muscle Cells

Chromatin is composed mostly of DNA and protein and is regulated by epigenetic mechanisms [57]. Epigenetic regulation can alter gene expression during transcription, post-transcription, translation, and post-translation. Thus, DNA, DNA-binding proteins, and histones can be modified by epigenetic mechanisms [6]. In the sections below, we discuss three major epigenetic regulations: DNA methylation, histone modification, and ncRNAs.

### 3.1. DNA Methylation

DNA methylation is processed by a group of DNA methyltransferase (DNMTs), such as DNMT1, DNMT3a, and DNMT3b. DNA methylation occurs when a methyl group is added to cytosine in cytosine phosphate guanine (CpG) islands. Transcriptional repressors bind with the methylated CpGs and hinder gene transcription. Several reports have demonstrated that DNA methylation plays a critical role in various diseases including atherosclerosis [58,59], vascular remodeling [60], and cardiovascular diseases [61]. For example, Dunn et al. showed that inhibiting DNA methylation could attenuate atherosclerosis by ameliorating endothelial dysfunction and restoring the mechanosensitive endothelial gene expression induced by oscillatory shear stress [62]. VSMCs can undergo phenotype switching from the contractile phenotype to the synthetic phenotype by environmental causes such as stress. Such phenotype switching is manifest as increased matrix stiffness and inflammation. Recently, the effects of matrix stiffness in VSMC phenotype and function were reported. Xie et al. reported that DNMT1 is reduced in arterial stiffness models such as acute aortic injury, the chronic kidney failure model, and calcified atherosclerotic lesions in human carotid arteries [63]. Inhibition of DNMT1 facilitates arterial stiffening in vivo and promotes osteogenic transdifferentiation and cellular stiffening of VSMCs in vitro. Mechanistically, DNMT1 elicits effects on contractility by regulating the promoter activities of smooth muscle α-actin and SM22α. Zhang et al. reported that DNMT3 is a direct target of *miR-143*, and DNMT3a regulates the hypermethylation of *miR-143* promoter in homocysteine-induced VSMC proliferation [64]. These results show that DNA methylation plays important roles in vascular smooth muscle remodeling. 

The role of DNA methylation in VC has been extensively investigated. Montes de Oca et al. found that DNMT activity and methylation of the promoter region of the SMC-specific protein SM22α are increased in high-phosphate-induced calcified VSMCs [65]. Azechi et al. [66] and Xie et al. [63] also demonstrated that downregulating DNMT1 expression facilitates Pi-induced arterial calcification by upregulating ALP expression and downregulating the DNA methylation level of the ALP promoter region. Lin et al. showed that *miR-204* is decreased in human aortic VSMCs with high Pi treatment and that it acts as a negative regulator during the process of osteoblastic differentiation [67]. Additionally, the methylation level of *miR-204* is regulated by DNMT3a. DNMT3a was revealed as a direct target of *miR-204* and to be involved in the process of human aortic SMCs differentiation. These results suggest that arterial calcification is regulated by the *miR-204*/DNMT3a regulatory circuit. Another study reported that downregulation of *miR-34b* in calcified VSMCs and arteries is associated with hypermethylation of CpG sites in the upstream region of *miR-34b* DNA regulated by DNMT3a. Lin et al. showed that *miR-34b* targets Notch1 in the regulation of VC [68]. Recently, da Silva et al. found that treatment of human aortic SMCs with calcium and phosphate increases DNMT3b, resulting in hypermethylation of osterix and bone sialoprotein (BSP), which ultimately leads to VC [69] (Table 1).

### 3.2. Histone Post-Translational Modification

The post-translational modifications (PTMs) of histones are important epigenetic regulatory mechanisms because the gene expression patterns can be inherited. PTMs can occur in histone proteins, which are located inside the nucleosome of chromatin. The N-terminal tails of histones are unstructured and can be modified [70]. The types of modifications that can occur in histone tails include acetylation, ADP-ribosylation, deimination, isomerization, methylation, phosphorylation, ubiquitination, and sumoylation [71]. Histone modifications have vital roles in transcriptional regulation, such as DNA replication, alternative splicing [72], DNA repair [73], and chromosome condensation [67]. Furthermore, not only histones but also other multiple proteins can undergo PTMs, resulting in changes in biological and pathological functions. In particular, the lysine residues of proteins undergo various PTMs such as methylation, ubiquitination, sumoylation, and acetylation. Lysine acetylation is a reversible PTM that can be conserved among different species. Lysine acetylation alters the charge on lysine residues and modifies the protein structure, thereby influencing enzyme activity, DNA-binding affinity, and protein stability [74,75].

### 3.3. Noncoding RNA

NcRNAs are functional RNAs that generally do not translate protein, and they are variable in length and structure. In addition to microRNAs (miRNAs), one of the extensively investigated ncRNAs, long noncoding RNAs (lncRNAs) and circular RNAs (circRNAs) have recently emerged as epigenetic regulators [76,77]. MiRNAs are single-stranded, endogenous, small noncoding RNAs of 16–25 nucleotides in length. They bind to the 3′-untranslated region of target mRNAs and mediate the translational repression or degradation of target transcripts. They play key roles in cellular processes such as differentiation, proliferation, migration, and apoptosis [78]. LncRNAs are longer than 200 nucleotides which are mostly transcribed by RNA polymerase II. They are spliced, capped at the 5′ end, and polyadenylated at the 3′ end [79]. Function of lncRNAs can vary depending on their localization. Nuclear lncRNAs can regulate gene transcription, whereas cytoplasmic lncRNAs may act as miRNA sponges [80]. CircRNAs are mainly produced by back-splicing reaction and are variable in length. CircRNAs are usually localized in cytoplasm and work as miRNA sponges [81]. To date, many studies have reported that ncRNAs act as both positive and negative regulators of the progression of VC.

Several reports demonstrated that ncRNAs positively regulate VC. For example, Gui et al. reported that *miR-135a**, *miR-712**, *miR-714*, and *miR-762* promote matrix mineralization in Ca/Pi-stimulated VSMCs by inhibiting calcium efflux proteins, NCX1, PMCA1, and NCKX4 [82]. In addition, Xia et al. revealed that *miR-2861* and *miR-3960* increase matrix mineralization in β-glycerophosphate-stimulated VSMCs by suppressing HDAC5 and Hoxa2 [83]. Moreover, Panizo et al. found that *miR-29b* enhances VC and downregulates the osteoblast differentiation inhibitors ACVR2A, CTNNBIP1, and HDAC4 in Ca/Pi-stimulated VSMCs [84]. Liu et al. revealed that *miR-32* inhibits PTEN through the activation of PI3K signaling, thereby increasing RUNX2 in β-glycerophosphate-stimulated VSMCs [85]. LncRNA-ES3 was reported to increase expression of calcification-related genes and BMF by acting as a *miR-34c-5p* sponge in high-glucose-stimulated human aorta VSMCs [86].

On the other hand, ncRNAs can also act as negative regulators of VC. *miR-125b* was revealed to downregulate osterix in osteogenic medium-stimulated HCASMCs [87]. Chao et al. showed that serum *miR-125b* can be used to predict the severity of VC [88]. *miR-204* was reported to inhibit RUNX2 in β-glycerophosphate-stimulated VSMCs [89]. Yanagawa et al. reported that *miR-141* inhibits BMP-2 in TGF-β-induced valvular interstitial cells (VICs) [90]. Additionally, *miR-30b* and *miR-30c* were revealed to inhibit RUNX2 in BMP-2 or β-glycerophosphate-stimulated VSMCs [91]. Jeong et al. reported that lncRNA Lrrc75a-as1 inhibits osteoblast-related genes in Pi-stimulated RVSMCs [92]. Circ-Samd4a was revealed to reduce VC by acting as miRNA sponges and regulating Camsap2 and Flna in Pi-stimulated RVSMCs [93]. These studies imply that discovering novel ncRNAs that control VC is important and that modulating these ncRNAs can be considered in the prevention and treatment of VC.

## 4. Role of Histone Modification in Vascular Calcification

Lysine acetylation of proteins is catalyzed by lysine acetyltransferases (KATs), which transfer the acetyl group of acetyl-coA to the ε-amino group of a lysine residue [94,95]. The reverse reaction is catalyzed by lysine deacetylases (KDACs), which comprise histone deacetylases (HDACs) and sirtuins (class III HDACs). Typically, acetylation of lysine in the tails of histones in the nucleosome weakens the interaction of these histones in the DNA by neutralization and reduction of the positive charge of the lysine residue, which then induces the activation of gene transcription [96]. Conversely, histone deacetylation alters the electrostatic properties of chromatin in a manner that favors the repression of gene transcription [97].

### 4.1. Histone Acetyltransferase in Vascular Calcification

Mammalian KATs can be categorized into two types, type A and type B, depending on their cellular localization. Type A KATs are nuclear KATs, whereas type B are cytoplasmic KATs [98]. Type A KATs mainly regulate the transcription of genes and are divided into five subgroups: MYST, GANT, p300/CBP, nuclear receptor coactivator family, and basal transcription factors [99,100]. Compared with type A KATs, only a few type B KATs have been reported, including KAT1 and KAT4 [101]. Type B KATs acetylate free histones and cytoplasmic substrates. The acetylation of lysines modifies the function of the protein by altering their structure or affinity to other binding partners. Thus, the acetylation of lysines regulates a variety of diseases, including neurological disorders [102], cancer [103], and CVDs [104]. Of the KATs, p300 has been extensively studied. p300 is a transcriptional coactivator that regulates gene expression by scaffolding, bridging, or activating intrinsic KAT [105]. 

Sierra et al. reported that p300 binds to RUNX2, resulting in an increase in osteocalcin promoter activity to induce osteogenesis in osteoblasts [106]. In aortic valvular calcification models, activation of p300 increases the acetylation of histones (H3 and H4) or RUNX2, thereby upregulating osteoblast-related genes such as osteocalcin and ALP. By contrast, C646, the most potent inhibitor of p300, reduces aortic valve calcification by suppressing the acetylation of H3 and H4. C646 is a promising anticancer agent owing to its ability to promote terminal differentiation and to induce growth arrest in cancer [107,108]. This suggests that C646 can also be used for the treatment of VC.

### 4.2. Histone Deacetylase in Vascular Calcification

Histone deacetylation is mediated by a family of HDACs that consists of 18 different HDAC molecules. Depending on their structural similarities and substrates, those 18 HDACs are divided into four different classes. HDAC1, 2, 3, and 8 belong to class I, whereas HDAC4, 5, 6, 7, 9, and 10 are class II. SIRT 1-7 (also known as HDAC12-18), class III HDACs, differ from the other HDACs in that their activities are dependent on NAD+. Lastly, there is only one HDAC in class IV, which is HDAC11 [109]. The deacetylase domain is conserved in these HDACs [110]. It is noteworthy that the HDACs themselves can undergo diverse PTMs including phosphorylation, ubiquitination, and even acetylation [109]. HDACs have been revealed to regulate multiple pathological processes, including cancer [111,112,113], apoptosis [114], inflammation [115], VC [116], and CVD processes [117]. Our group also reported the role of HDACs in the development of cardiac hypertrophy and heart failure [118,119,120,121,122,123]. Although the roles of HDACs in VC have not been extensively investigated, it is likely that HDACs work as transcriptional repressors of their targets and that the overall effects of those HDACs may depend on the properties of their target genes.

#### 4.2.1. Class I HDACs

HDAC1, 2, and 8 are mainly localized in the nucleus. On the other hand, HDAC3 is sometimes detected in the cytoplasm (Figure 1). Although class I HDACs have short regulatory domains compared with those of the class IIa HDACs, class I HDACs are more actively modified than are the other classes of HDACs. Among the class I HDACs, HDAC1 and HDAC2 have similar sequences and structures [109]. Many studies have revealed that HDACs are correlated with the development of CVD or CVD risk factors, including hypertension, cardiomyopathy [118,122,124], myocardial infarction [125], and heart failure [126,127]. Among the class I HDACs, we discovered that HDAC1 is involved in the development of VC in both high-phosphate-treated VSMCs and vitamin D_3_-injected mice [116]. In diverse VC models, the degradation of HDAC1 protein is increased and protein amount is significantly reduced. HDAC1 K74 is a ubiquitination target residue, and polyubiquitination of K74 mediates the VC-induced protein degradation. We also found that MDM2 E3 ligase is involved in HDAC1 polyubiquitination of VC; overexpression of MDM2 promotes VC, whereas knockdown of MDM2 inhibits VC because degradation of HDAC1 is suppressed. In addition, we observed that MDM2 is upregulated in the intimal and medial layers of calcified human coronary arteries. It is noteworthy that RG 7112, an MDM2 inhibitor, completely inhibits VC. These results suggested that the MDM2/HDAC1 axis is a novel signaling pathway that regulates VC in VSMCs, and that targeting this axis can be considered to prevent VC [116] (Table 2). Like C646, considering that RG 7112 is under extensive investigation as an anticancer drug [128,129], the extension of its use to VC would be of great interest.

#### 4.2.2. Class II HDACs

The class II HDACs can be divided into two subclasses: class IIa (HDAC4, 5, 7, and 9) and class IIb (HDAC6 and HDAC10). The class IIa HDACs are characterized by a long N-terminal domain extension in addition to their catalytic domain, whereas the class IIb HDACs such as HDAC6 and HDAC10 contain two catalytic domains [134]. The class IIa HDACs differ from the other HDACs because they contain binding sites for myocyte enhancer factor 2 (MEF2), a nuclear localization signal in the N-terminal domain and a nuclear export signal in the C-terminal domain (Figure 1). The expression of the class II HDACs is more tissue-specific compared with that of the other classes of HDACs. Like the class I HDACs, the functions of the class II HDACs are also regulated by PTMs, including phosphorylation, acetylation, ubiquitination, and sumoylation. Class II HDACs have crucial roles in the development of the heart, vasculature, muscle, bone, brain, and immune system [135].

HDAC4 is a key regulator of cell growth [136], differentiation [137], proliferation [138], and migration [139] in various cell types and participates in physiologic and pathologic processes such as proliferation and migration during neointimal hyperplasia [138]. Recently, Zheng et al. reported that suppression of HDAC4 inhibits the proliferation of VSMCs induced by platelet-derived growth factor-BB [140]. They demonstrated that HDAC4 plays a key role in cellular processes. Ren et al. found that HDAC4 is required for the proliferation and migration of VSMCs to regulate neointimal hyperplasia. In addition, HDAC4 was shown to have important functions in bone formation and cartilage development. Whole-body deletion of HDAC4 by conventional knocking out in mice results in premature ossification as a result of ectopic hypertrophy of chondrocytes [141]. Moreover, double knockout of HDAC4 and RUNX2 causes low bone mass. Thus, HDAC4 is critical in bone formation. Since VC and bone formation share similar features, one can easily assume that HDAC4 may regulate VC as well. Recently, Abend et al. reported that HDAC4 is upregulated early in VC, which significantly increases the expression of RUNX2 and OPN, the markers of VC [130]. Knocking-down of HDAC4 in VSMCs reduces VC. Class IIa HDAC4 can be exported from the nucleus in response to activation of stimulus-dependent kinase. Likewise, the salt-inducible kinase (SIK)-mediated phosphorylation of HDAC4 keeps HDAC4 in the cytoplasm. In the cytoplasm, HDAC4 binds with adaptor protein ENIGMA (Pdlim7) to promote VC. Conversely, pan-SIK inhibitors translocate HDAC4 from the cytoplasm to the nucleus, which results in the inhibition of VC (Table 2).

HDAC5 undergoes nuclear-cytoplasmic shuttling and acts as an important regulator in cellular and epigenetic processes that underlie the progression of human disease, including cardiac diseases [142,143,144,145], tumorigenesis [146], and VC [83]. Xia et al. reported that HDAC5, a direct target of *miR-2861*, inhibits RUNX2 expression in β-glycerophosphate-induced osteogenic transdifferentiation of VSMCs [83]. Several studies revealed that HDAC4 or HDAC5, class II HDACs, promote Smad ubiquitin regulatory factor 1-mediated degradation of RUNX2, which induces BMP-2-mediated osteoblast differentiation and bone formation [147,148]. We also identified that miR-mediated inhibition of HDAC5 can induce VC by decreasing the inhibition of RUNX2 in high-phosphate-stimulated VSMCs (unpublished data). These studies show that the pathogenesis of VC is similar to the process of bone formation.

HDAC9, as a member of the HDAC IIa family, regulates diverse diseases including neurodegenerative diseases [149,150], cancer [151,152], and cardiovascular diseases [153,154]. Recently, Malhotra et al. revealed that HDAC9 is associated with abdominal aortic calcification using genome-wide association meta-analysis [131]. HDAC9 was increased in osteogenic medium-induced human aortic SMCs, which increased RUNX2 expression and reduced contractility. Using siRNA, knocking-down of HDAC9 significantly reduced VC by inhibiting RUNX2. Furthermore, HDAC9 and MGP KO mice (HDAC9^−/−^ and MGP^−/−^) show reduced VC compared with MGP KO mice (MGP^−/−^). These results demonstrate that HDAC9 may function as an activator of atherosclerosis and VC. However, further studies need to be considered to elucidate the precise mechanism of HDAC9 in the regulation of VC.

Several reports revealed that HDAC6, a member of the HDAC IIb subfamily, regulates the progression of diverse diseases including cancer [155], bone remodeling [156], and vascular remodeling [157]. Fu et al. showed that HDAC6 was significantly reduced in human calcified aortic valves, and knocking-down of HDAC6 increased RUNX2 expression and induced aortic calcification in VICs [132]. Rao and colleagues found that HDAC6, the regulator of endoplasmic reticulum stress in breast cancer cells, inhibits endoplasmic reticulum stress through deacetylation of glucose-related protein 78 (GRP 78) and suppression of activating transcription factor 4 (ATF4)-mediated osteogenesis [158]. Previous studies reported that endoplasmic reticulum stress is related to aortic valve calcification through ATF signaling [159,160]. ATF4 is a sensor of endoplasmic reticulum stress and also works as a transcription factor that involves the osteogenic pathway [161]. According to previous reports, it is likely that pulmonary arterial hypertension, an elevation of pulmonary arterial pressure due to heart, lung, or systemic disorders [162], may be related to VC [163,164]. Ruffenach et al. claimed that VC may be associated with pulmonary artery stiffness in patients with pulmonary arterial hypertension [163]. Boucherat et al. recently reported that expression of HDAC6 protein is increased in response to upregulated HSP90 in pulmonary artery SMCs isolated from patients with pulmonary arterial hypertension [165]. Hsp90 is a central coordinator involved in many diverse cellular pathways upon stress [166]. Weisell et al. reported that Hsp90 is downregulated in patients with calcified aortic valve diseases [167]. Taken together, these results suggest that HDAC6 may regulate VC in aortic valve disease and pulmonary arterial hypertension.

#### 4.2.3. Class III HDACs

Compared with class I and II HDACs, sirtuins (SIRTs), class III HDACs, are nicotinamide adenine dinucleotide (NAD+)-dependent deacetylating enzymes that were first identified in yeast. Sirtuins have highly conserved NAD+-binding domains with an ~250 amino acid core and variable amino- and carboxy-terminal extensions. Mammals possess seven sirtuins with different properties of subcellular localization, enzymatic activity, and binding targets. SIRT1 and SIRT6 are mainly localized in the nucleus, but they can translocate into the cytoplasm under specific conditions [168]. SIRT2 resides mostly in the cytoplasm, whereas SIRT3, SIRT4, and SIRT5 are localized in mitochondria. In contrast, SIRT7 is localized in the nucleus (Figure 1). SIRTs regulate essential molecular pathways in eubacteria, archaea, and eukaryotes and are involved in stress resistance, apoptosis, aging, senescence, and inflammation. SIRT1 and SIRT6 are characterized for their protective roles against inflammation, vascular aging, heart disease, and atherosclerotic plaque development [169]. SIRT1 antagonizes p53-induced cellular senescence in mouse embryo fibroblast (MEF) and suppresses oxidative stress-mediated premature senescence in vascular endothelial cells [170]. Senescent cells are also detected in Pi-induced VSMCs and in the calcified areas of an adenine-induced chronic renal failure rat model [133,171]. Therefore, it seems that cellular senescence is positively correlated with VC. Also, Takemura et al. reported that high-phosphate-induced senescence is related to downregulation of SIRT1 expression, leading to p21 activation. Further evidence showed that knocking-down of SIRT1 accelerates the high-phosphate-induced cellular senescence and VC in SMCs [133,171]. These results suggest that SIRT1 induces arterial calcification by inhibiting osteoblastic transdifferentiation such as RUNX2 expression in conditions such as aging and diabetes (Table 2).

## 5. Crosstalk between Osteogenic Transcription Factors and HDACs in Noncalcified Smooth Muscle Cells

Since the mechanism of VC is similar to the process of osteoblast differentiation, for a better understanding of the roles of HDACs in VC, in this section, we review studies of HDACs in ossification other than in SMCs. Osteogenic transcription factors such as RUNX2, MSX2, OPG, and OPN are expressed in multiple tissues and regulate the progression of diverse diseases such as cancer [172,173], fibrosis [174,175], and bone diseases [176]. Moreover, HDACs act as important epigenetic regulators in various diseases and play a critical role in cellular processes [109]. To date, many studies have reported that there is crosstalk between HDACs and other osteogenic transcription factors in various cells. Manzotti et al. revealed that HDAC1 and HDAC6 activate RUNX2 transcription in cancer cells, such as thyroid TPC1 cells and MDA-MB231 cells by binding to the RUNX2 P1 promoter. In particular, HDAC6 interacts with RUNX2 to regulate downstream targets such as TERF1 and platelet-derived growth factor β in thyroid TPC1 cells [155]. In addition, Huang et al. reported that both HDAC1 and HDAC2 are required for RUNX2 to increase TGF-β-induced OPN expression, which promotes epithelial-to-mesenchymal transition-mediated invasion in nonsmall cell lung cancer cells [177]. Among the class I HDACs, HDAC1 and HDAC3 interact with RUNX2, respectively, during osteoblast differentiation and inhibit RUNX2-mediated osteogenesis, including osteocalcin in osteogenic cells like MC3T3-E1, ROS12/2.8, and primary bone marrow cells [178,179]. In addition, Yoshizawa et al. reported that MSX2 interacts with RUNX2/Osf2 and prevents ossification by recruiting HDAC1 in mesenchymal progenitor C3H10T1/2 fibroblastic cells [180]. In studies of class II HDACs, *miR-29b*-mediated suppression of HDAC4 was shown to promote osteoblast differentiation by activating RUNX2 in MC3T3-E1 cells [181]. Another study conducted by Westendorf et al. revealed that the binding between HDAC6 and RUNX2 in the nucleus induces osteoblast differentiation [156]. On the other hand, the interaction between HDAC7 and RUNX2 suppresses osteoblast maturation by inhibiting RUNX2 activity in osteoblast lineage cells such as multipotent progenitor C2C12 cells [182].

## 6. HDAC Inhibitor/Activator and Therapeutic Application in Vascular Calcification

Since diverse diseases are regulated by HDACs, HDAC inhibitors may be potential therapeutic agents. Various HDAC inhibitors have been evaluated as therapeutic agents for myelodysplastic syndromes, advanced leukemia, neurologic diseases, immune disorders, and CVDs [98,183]. To date, six HDAC inhibitors such as vorinostat, belinostat, romidepsin, sodium butyrate, valproic acid (VPA), and panobinostat are approved by the Food and Drug Administration as drugs for diverse diseases. However, studies of HDAC inhibitors in VC and CVDs are still in preclinical phases. 

Most HDAC inhibitors act as pan-HDAC inhibitors that block more than two HDACs. HDAC inhibitors can be classified into four different classes: cyclic peptide (e.g., apicidin), hydroxamic acid (e.g., vorinostat and trichostatin A), short-chain fatty acid (e.g., VPA), and benzamide (e.g., entinostat and sirtinol). Other novel classified chemical classes are as follows: isothiocyanate (e.g., sulforaphane) and stilbenoid (e.g., resveratrol) (Figure 2). In general, it is widely accepted that HDAC inhibitors may be used in the treatment of epilepsy, angiogenesis, cardiac hypertrophy, heart failure, fibrosis, and myocardial infarction [98]. Some HDAC inhibitors including apicidin, trichostatin A, vorinostat, tubacin, and sirtinol are reported to aggravate VC [107,116,132,171,184], whereas other inhibitors such as entinostat, VPA, and sulforaphane have shown potential as therapeutic agents in VC [185,186,187]. Since studies of these HDAC inhibitors in VC are at the preclinical level (Table 3), further studies are necessary to determine their use as therapeutic agents.

### 6.1. HDAC Regulators that May Exaggerate VC

Trichostatin A, which is a pan-HDAC inhibitor, and apicidin, which is a class I HDAC inhibitor, increase VC by activating RUNX2 in Pi-induced VSMCs [116,184]. Additionally, TSA enhances VC in a vitamin D_3_-induced in vivo VC model. As mentioned above, the E3 ligase MDM2 is dramatically increased after the induction of VC, which enhances polyubiquitination and results in the subsequent degradation of HDAC1. The loss of HDAC1 activity plays a crucial role in the progression of VC. Hence, HDAC inhibitors may accelerate VC in vivo [116].

Vorinostat (also known as SAHA) is as a pan-HDAC inhibitor that is widely used as a treatment for epilepsy, cancer, and other diseases [189,190]. Vorinostat is reported to increase osteogenic differentiation through H4 acetylation and modulation of insulin/Akt/FoxO1 signaling in osteoblast cells [191]. Additionally, vorinostat increases acetylation of H3 and H4, which increases RUNX2 and ALP in high-calcium/phosphate-induced porcine aortic VICs (pVICs) [107].

Tubacin is an HDAC6 inhibitor that was reported to promote aortic valvular calcification by activating GRP78 acetylation in the endoplasmic reticulum. GRP78 acetylation induces osteogenic differentiation signaling, including ATF4 and RUNX2 expression in osteogenic medium-induced human aortic VICs (hVICs) [132]. This study demonstrated that tubacin has a procalcification role and suppresses HDAC6. 

Sirtinol, an inhibitor of Sirt1 and Sirt2, reduces DNA replication and transcription in the hepatitis B virus. Recently, Takemura et al. reported the effect of sirtinol, which promotes high-glucose-induced VC by activating the expression of RUNX2 and OCN [133]. Moreover, sirtinol also increases mineralization as shown in alizarin red S staining of osteogenic-medium-induced SMCs [171].

### 6.2. HDAC Modulators that Can Block VC

Entinostat (also known as MS275) is a potent class I and IV HDAC inhibitor. It is currently in clinical trials for the treatment of cancers and has therapeutic effects on bone defects [192] and cardiovascular diseases including hypertension [193,194], cardiac hypertrophy [195], and heart failure [196]. In the osteogenic-medium-induced calcification models, Li et al. reported that entinostat reduces RUNX2 expression and ALP activity by downregulating Wnt signaling, which interacts with GSK-3β, β-catenin, and p-SMAD1/5/8 in calcified VICs [185].

VPA, a class I HDAC inhibitor, is used for the treatment of epilepsy and is in clinical trials for cancer [197]. Dai et al. showed that calcification stress-induced reactive oxygen species products potentiate autophagy. Interestingly, VPA inhibits calcium deposition by upregulating autophagy in phosphate-induced VSMCs and in the aortic walls of adenine-induced chronic kidney disease rats [186]. This result was the opposite of the previously reported phenomenon that inhibition of autophagy reduces VC. The authors explained that this finding may be caused by an increase in release of matrix vesicles. Thus, this study demonstrated that VPA acts as an inducer of autophagy and counteracts calcification by inhibiting matrix vesicle release [186].

Sulforaphane is an isothiocyanate derived from broccoli [198] that acts as an inhibitor of class I HDACs and HDAC6 in diverse cell types such as HCT116 [199,200], prostate cell lines [201], and human peripheral blood mononuclear cells [202]. Recently, Zhang et al. reported that sulforaphane decreased reactive oxygen species production by increasing nuclear factor-erythroid2-related factor-2 (NRF-2) expression and subsequently inhibiting RUNX2 expression in β-glycerophosphate-induced rat VSMCs [187]. These results demonstrated that sulforaphane may have an anticalcification role in oxidative stress-induced VC.

Resveratrol [203] is a polyphenol phytoalexin that is present in plants and that works as a SIRT1 (also known as HDAC 12) activator. It may prevent the progression of diverse diseases including neurodegenerative diseases, cancer, and cardiovascular diseases [204,205]. Takemura et al. reported that resveratrol also has protective effects on calcification by decreasing osteogenesis and has antioxidative effects by increasing NRF2 mRNA levels in a hyperphosphatemia-induced calcification model [133]. NRF2 has a protective role on calcification by counteraction of inflammation through the inhibition of oxidative stress and pro-inflammatory cytokines [206]. Many researchers have reported that oxidative stress can activate osteogenic signals such as RUNX2 in VSMCs [21]. Zhang et al. found that resveratrol effectively preserves oxidative injury by inhibiting VC through the regulation of SIRT-1 and NRF2 in rat aortic VSMCs [188]. However, further studies may be necessary to understand the regulatory mechanism of SIRT1 and NRF2 in VC.

## 7. Conclusions

VC is an interactive and complex process involving multiple factors such as pro- and anti-calcifying factors, apoptosis, mitochondrial dysfunction, cellular senescence, and endoMT. Since VC is an important risk factor for CVD, investigation of novel regulators may be needed to resolve VC. In the current review, we summarized and discussed the roles of epigenetic regulators including DNA methylation, HDACs, and ncRNAs and proposed therapeutic applications of the HDACs in the treatment of VC. DNA methylation and ncRNAs including miRNAs, lncRNAs, and circRNAs can act as positive or negative regulator of VC and may serve as novel therapeutic targets. Furthermore, these epigenetic regulators may crosstalk with each other to regulate VC. In addition, HDACs have several different classes and many subtypes, and their action may vary in the development of VC. Although some HDACs play pro-calcification roles, others demonstrate anti-calcification effects. Since the mechanism of action of HDAC inhibitors and activators may differ depending on the type of HDACs, further studies are required to understand the precise role of individual HDACs and to validate the effects of HDAC inhibitors or activators and to determine the optimal route of medication administration. Here, we summarized the roles of many epigenetic regulators in the development of VC. Although multiple complex mechanisms await further extensive investigations, this review will add new insights for the development of new therapeutics against VC.

## Figures and Tables

**Figure 1 ijms-21-03246-f001:**
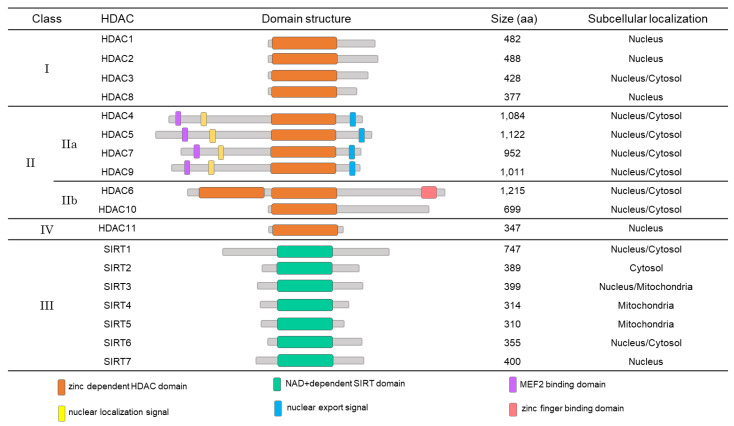
Classification and domain structure of the histone deacetylases (HDACs). The colored boxes on the domain structure are indicated as follows: orange, zinc-dependent HDAC domain; mint, NAD^+^-dependent SIRT domain; purple, MEF2 binding domain; yellow, nuclear localization signal; blue, nuclear export signal; and peach, zinc finger binding domain. HDAC, histone deacetylase, SIRT, sirtuin; and MEF2, myocyte enhancer factor 2.

**Figure 2 ijms-21-03246-f002:**
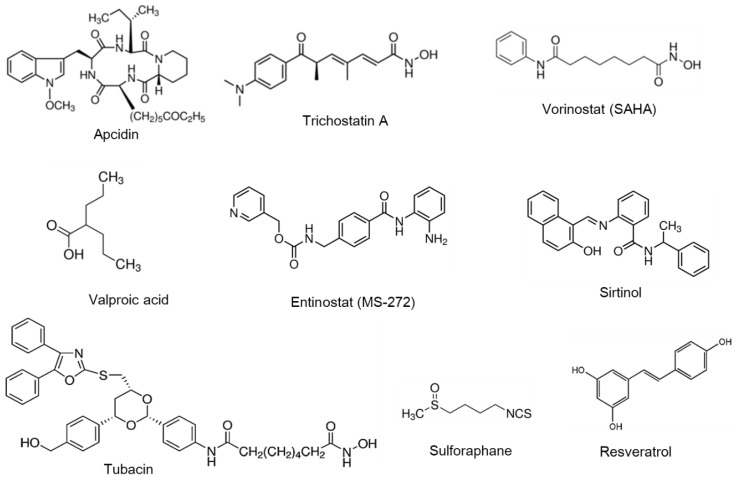
Structure of HDAC inhibitors and activators. The chemical structures of the HDAC inhibitors and activators discussed in this review are depicted.

**Table 1 ijms-21-03246-t001:** Role of DNA methyltransferases (DMNTs) in vascular calcification.

Type of DNMTs	Target Gene	Role in VC	References
DNMT1	ALP promoter	Inhibit	[66]
DNMT3a	*miR-204* & *miR-34b*	Promote	[67,68]
DNMT3b	Osterix and BSP	Promote	[69]

**Table 2 ijms-21-03246-t002:** Role of histone deacetylases (HDACs) in vascular calcification.

Type of Class	HDACs	Expression	PTMs	Regulator	Function	Reference
Class I	HDAC1	Downregulation	Ubiquitination	MDM2	Decrease Runx2	[116]
	HDAC2	N.C.				
	HDAC3	N.C.				
	HDAC8	N.C.				
Class IIa	HDAC4	Upregulation	Phosphorylation	SIK and Pdlim7	Increase Runx2 and OPN	[130]
	HDAC5	Downregulation		*miR-2861*	Decrease Runx2	[83]
	HDAC7	N.C.				
	HDAC9	Upregulation			Increase Runx2	[131]
Class IIb	HDAC6	Downregulation		ATF4	Decrease Runx2	[132]
	HDAC10	N.D.				
Class III	SIRT1	Downregulation		p21	Decrease Runx2	[133]
Class IV	HDAC11	N.D.				

Abbreviation: N.C., not changed; N.D., not detectable; MDM2, Mouse double minute 2 homolog; PTM, post-translational modification; SIK, salt-inducible kinase; and Pdlim7, protein ENIGMA.

**Table 3 ijms-21-03246-t003:** Characteristics of HDAC inhibitors and activators in vascular calcification.

HDAC Modulator	Chemical Classification	HDAC Specificity	Effect on VC	Study Model	Cell Type	Mechanism	References
*HDAC inhibitor*							
Apicidin	Cyclic peptide	Class I	Promote	2 mM Pi	RVSMCs	Reduces HDAC1 and increases Runx2	[116]
Trichostatin A	Hydroxamic acid	Class I, II, IV	Promote	3 mM Pi	HASMCs	Reduces HDAC1 and increases Runx2	[184]
Vorinostat	Hydroxamic acid	Class I, II, IV	Promote	1.5 mM Ca^2+^ and 2 mM Pi	pVICs	Enhances H4 acetylation, Runx2, and OPN	[107]
Tubacin	Hydroxamic acid	HDAC6	Promote	10 mM β-GP, 10 nM DM, 4 μg/mL vitamin D_3_, and 8 mM CaCl_2_	hVICs	Activates ER stress and increases Runx2	[132]
Sirtinol	Benzamide	SIRT1, SIRT2	Promote	5 mM β-GP and 2.6 mM CaCl_2_	HCASMCs	Increase senescence and promotes Runx2 and osteocalcin	[171]
Entinostat	Benzamide	Class I, IV	Inhibit	10 nM DM, 10 mM 10 mM β-GP, and 50 mg/mL AA	pVICs	Reduce Wnt signaling, Runx2, and ALP	[185]
Valproic acid	Short-chain fatty acids	Class I, IIa	Inhibit	3mM Pi	BASMCs	Enhances autophagy and reduces Runx expression	[186]
Sulforaphane	Isothiocyanate	Class III	Inhibit	770 mg β-GP, 11 mg L-AA, and 10 nM DM	RVSMCs	Reduces oxidative stress and Runx2	[187]
*HDAC activator*							
Resveratrol	Stilbenoid	SIRT1, SIRT2	Inhibit	3.2mM Pi or 770 mg β-GP, 11 mg L-AA, and 10 nM DM	HASMCs or RVSMCs	Reduces senescence and Runx2 and increases Nrf2	[133,188]

Pi, inorganic phosphate; β-GP, β-glycerophosphate; DM, dexamethasone; AA, ascorbic acid; RVSMCs, rat vascular smooth muscle cells; HASMCs, human aortic smooth muscle cells; pVICs, porcine aortic valvular interstitial cells; hVICs, human aortic valvular interstitial cells; HCASMCs, human coronary artery smooth muscle cells; BASMCs, bovine aortic smooth muscle cells; and Nrf2, nuclear factor-erythroid related factor 2.

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
