# Peer review of "Roles of Histone Acetylation Modifiers and Other Epigenetic Regulators in Vascular Calcification"

_ijms, 2020, doi:10.3390/ijms21093246_

Round 1

Reviewer 1 Report

To authors:

Kwon et. al. reviewed roles of histone acetylation in vascular calcification and therapeutic possibilities of the modifiers. Authors demonstrated recent findings regarding epigenetic mechanisms of vascular calcification including DNA methylation, histone modification, and micro RNA with suitable references. They focused on the roles of HDAC in vascular calcification, and introduced previous findings of HDAC activators and inhibitors. They explained the characteristics of these mediators and some of them such as Entinostat, Sulforaphane, and Resveratrol, are promising for clinical use, although future studies are needed. The review seems to be beneficial for researchers focusing on this area.

Minor: Please complete the sentence in Line 450-451.

Author Response

Reviewer #1

We appreciate reviewer #1’s positive comments

As the reviewer suggested, we revised the sentence in Line 450-451 (currently 475~478) as highlighted.

Reviewer 2 Report

In the present review, the authors highlight the role of vascular calcification in different cardiovascular diseases. In particular, they focus attention on DNA methylation and histone modification and microRNAs. They summarize the current experimental evidence regarding the role of epigenetic regulators, including histone deacetylases.

The manuscript is attractive and well written, but some issues should be necessary addressed.

  • Section “2.3. Apoptosis, mitochondrial dysfunction, and cellular senescence” is very short and have only marginal information. It should be expanded or integrate inside another paragraph.

  • Can the authors report the pharmacological therapies already used to reduce the VC? I have found this sentence only in the abstract, but the text completely misses this information. Could be interesting to report pharmacological compounds that act modulating DNA methylation, histone modification or microRNAs.

  • The title seems more precise, but the review also reports other processes behind the histone acetylation. In my opinion, it should be revised.

  • The conclusion section should be expanded.

  • A concept that is completely missing in the text is the process of endo-MT, which seems most important for the calcification process. Please add a dedicated paragraph.

  • In the last years, long noncoding RNA are emerging as essential molecules involved in the regulation of various physiological and non-physiological processes, such as calcification. It should be interesting to report also a highlight on them.

Author Response

Reviewer #2

We appreciate reviewer #2’s suggestions and comments.

  1. Description on section 2.3.

Section “2.3. Apoptosis, mitochondrial dysfunction, and cellular senescence” is very short and have only marginal information. It should be expanded or integrate inside another paragraph.

We added more descriptions on other mechanisms at the section 2.3., as highlighted Line 117~142. For example, we added several sentences for the explanation on mitochondrial damage and microvesicles. More importantly, we added new paragraph regarding EndoMT. Accordingly, we also revised the subtitle from ‘Apoptosis, mitochondrial dysfunction, and cellular senescence’ to ‘other factors affecting VC’.

  1. Description on pharmacological therapies.

Can the authors report the pharmacological therapies already used to reduce the VC? I have found this sentence only in the abstract, but the text completely misses this information. Could be interesting to report pharmacological compounds that act modulating DNA methylation, histone modification or microRNAs.

To our knowledge, no specific pharmacologic drugs are available, although some reagents such as resveratrol or sulforaphane are under investigation as written in the review. Rather, in clinical situation, the indirect or non-specific interventions such as reduction of serum phosphate by dialysis or other approaches are used to reduce VC.

Likewise, we revised a sentence in the abstract

Previous version: Although pharmacologic therapies to treat VC are available, VC is commonly observed in patients with chronic kidney disease and CVD

Revised version: Although VC is commonly observed in patients with chronic kidney disease and CVD, specific drugs for VC have not been developed.

  1. Title

The title seems more precise, but the review also reports other processes behind the histone acetylation. In my opinion, it should be revised.

We revised the title as followings:

Previous title: Roles of histone acetylation modifiers in vascular calcification

Revised title: Roles of histone acetylation modifiers and other epigenetic regulators in vascular calcification

  1. Conclusion

The conclusion section should be expanded.

We added new sentences to the conclusion.

  1. EndoMT

 A concept that is completely missing in the text is the process of endo-MT, which seems most important for the calcification process. Please add a dedicated paragraph.

We added a brief review on EndoMT at the section of 2.3 (Line 136~142)

  1. Long non-coding RNA

In the last years, long noncoding RNA are emerging as essential molecules involved in the regulation of various physiological and non-physiological processes, such as calcification. It should be interesting to report also a highlight on them

 We added new paragraph regarding long non-coding RNA as well as circular RNA at the section 3.3 of previous miRNA (Line 201~236). In the revised manuscript, we also added our new results of lncRNA Lrrc75a-as1 and circRNA, circ-samd4a that were recently published (ref 92, 93).

Accordingly, we revised the title from miRNA to non-coding RNA and replaced miRNA with non-coding RNA in the entire manuscript.

Round 2

Reviewer 2 Report

I have no other comments.